

# Event-scale power law recession analysis: Quantifying methodological uncertainty

David N. Dralle[1], Nathaniel J. Karst[2], Kyriakos Charalampous[1,3], and Sally E. Thompson[1]

[1]University of California Berkeley, Berkeley, CA
[2]Babson College, Wellesley, MA
[3]University of Bristol, United Kingdom

*Correspondence to:* David N. Dralle (dralle@berkeley.edu)

**Abstract.**

The study of single streamflow recession events is receiving increasing attention following the presentation of novel theoretical explanations for the emergence of power-law forms of the recession relationship, and drivers of its variability. Individually characterizing streamflow recessions often involves describing the similarities and differences between model parameters fitted to each recession time series. Significant methodological sensitivity has been identified in the fitting and parameterization of models that describe populations of many recessions, but the dependence of estimated model parameters on methodological choices has not been evaluated for event-by-event forms of analysis. Here, we use daily streamflow data from 16 catchments in northern California and southern Oregon to investigate how combinations of commonly used streamflow recession definitions and fitting techniques impact parameter estimates of a widely-used power-law recession model. We show that: (i) methodological decisions, including ones that have received little attention in the literature, can impact parameter value estimates and model goodness-of-fit; (ii) the central tendencies of event-scale recession parameter probability distributions are largely robust to methodological choices, in the sense that differing methods rank catchments similarly according to the medians of these distributions; (iii) recession parameter distributions are method-dependent, but roughly catchment-independent, such that changing the choices made about a particular method affects a given parameter in similar ways across most catchments; and (iv) the observed correlative relationship between the power law recession scale parameter and catchment antecedent wetness varies depending on recession definition and fitting choices.

## 1 Introduction

Streamflow recession analysis has the goal of characterizing recession behavior in terms of phenomenological models of decreases in flow ($q$, with units of [L T$^{-1}$] or [L$^3$ T$^{-1}$]) over time, typically represented with a power-law differential equation (Boussinesq, 1877; Hall, 1968; Tallaksen, 1995):

$$\frac{dq}{dt} = -aq^b \implies q(t) = \left(q_0^{1-b} - (1-b)at\right)^{\frac{1}{1-b}}. \tag{1}$$



There is no universally agreed upon procedure for performing power law recession analysis, however most approaches are comprised of two key steps: (i) Identify and isolate periods of flow recession using the hydrograph and (optionally) other hydroclimatic datasets – a step referred to here as "recession extraction"; and (ii) Use the isolated periods of recession to parameterize the power law model – a step we refer to as "fitting".

Classical recession analysis performs the fitting step in a single operation: $[log(q), \log(-dq/dt)]$ point pairs are computed for multiple recession periods, and the recession parameters are then obtained from the slope and intercept of a line fitted to the $[\log(q), \log(-dq/dt)]$ point cloud (e.g. Brutsaert and Nieber, 1977; Stoelzle et al., 2013; Tague and Grant, 2004; Basso et al., 2015; Clark et al., 2009; Sawaske and Freyberg, 2014; Bogaart et al., 2016). This form of "lumped" recession analysis is empirically and theoretically motivated. Practically, it reasonably captures observed non-linearity in the hydrograph recession.

Theoretically, it uses a model form that is predicted by solutions of the hydraulic groundwater equations (Boussinesq, 1904; Troch et al., 2013). Lumped recession analysis has been used for inverse modeling, the development of flow separation algorithms, characterization of aquifer properties, and parameterization of hydrologic models, among other applications (Vogel and Kroll, 1992; Rupp and Selker, 2006a; Rupp et al., 2004; Szilagyi et al., 1998; Huyck et al., 2005; Bogaart et al., 2016; Tague and Grant, 2004).

Recently, several authors have attributed physical meaning to observed variability *across* individual recessions within a single catchment, triggering an increase *event-scale* recession analyses (Dralle et al., 2016; Ghosh et al., 2016; Ye et al., 2014; Wittenberg, 1999; Biswal and Marani, 2010, 2014; Biswal and Nagesh, 2014; Harman et al., 2009; Mutzner et al., 2013; Bart and Hope, 2014; Shaw, 2016; Dralle et al., 2015; Patnaik et al., 2015; Shaw and Riha, 2012; Vogel and Kroll, 1996; Chen and Krajewski, 2016). Whereas classical, lumped recession analysis seeks a single recession model parameterization to describe

all hydrograph recessions for an individual catchment, the goal of event-scale recession analysis is to interpret variations in catchment response to rainfall as a function of the properties of rainfall events (e.g. Harman et al., 2009) or the catchment state (e.g. Biswal and Marani, 2010; Shaw and Riha, 2012).

Among the many issues associated with event-scale analysis (Dralle et al., 2015), perhaps the most challenging are the numerous subjective choices needed to establish consistent criteria for recession identification and fitting (Westerberg and

McMillan, 2015). For lumped analyses, Brutsaert and Nieber (1977) established a derivative-based method, which avoids the issue of needing to determine the precise start day of a recession event. Event-scale analyses, however, must identify the start and end of each recession event and select one of many fitting techniques to obtain $(a, b)$ values.

Despite the growing number of event-scale recession studies, it remains unclear to what extent the particular method of recession extraction and fitting could alter features of the computed populations of recession parameters. If uncertainty due to

methodological choices exceeds physically-derived variations in the recession parameters, new and less ambiguous methods will be needed to allow empirical comparative analyses, and to test hypotheses derived from novel theories (e.g. Biswal and Marani, 2010; Clark et al., 2009; Harman et al., 2009). Previous work has demonstrated that method dependent variability in recession parameters in lumped analysis can be larger than natural variability between catchments (Stoelzle et al., 2013). For event-scale recession analysis, Chen and Krajewski (2016) demonstrate sensitivity of the recession exponent to recession

length and start time relative to a flow peak. However, no systematic study has been undertaken to examine sensitivity of both




$a$ and $b$ to some of the most common methodological choices made during event-scale power law recession analysis. Given the early stage of event-scale recession exploration, it is an opportune time to determine the methodological limitations associated with event-scale techniques, hopefully supporting inter-comparability and consistency in future work.

Analogously to Stoelzle et al. (2013) and Chen and Krajewski (2016), this study examines the sensitivity of recession pa-
rameter values to the various methodological choices to be made when performing event-scale recession extraction and fitting. Unlike lumped recession analysis (Vogel and Kroll, 1992; Brutsaert and Nieber, 1977; Kirchner, 2009), no set of canonical methods of event-scale recession analysis have been established. We propose breaking down the two steps of recession analysis – recession extraction and power law model parameterization – into four methodological choices; three concerning recession extraction, and one concerning model parameterization:

1. *The minimum allowable length of a recession event*

2. *The definition of the beginning of a recession event*

3. *The definition of the end of a recession event*

4. *The method of power law model fitting*

In this work, we select two end-member settings that define realistic methodological limits for each of the above four
choices. The resulting 16 combinations of method choices, as applied to a broad flow dataset, provide a basis for constraining method-dependent uncertainty in the populations of recession parameters.

## 2 Methods

### 2.1 Study sites

The analyses in this study are performed using United States Geologic Survey daily streamflow data for the set of 16 U.S.
catchments from northern California and southern Oregon summarized in Tab. 1. While recession analysis can be performed using more frequently sampled discharge data, we use daily data because it is the most common choice in event-scale recession literature. The study catchments are situated within the U.S. western coastal Mediterranean climate region, which is character-ized by a distinct rainy season, followed by a pronounced dry season during which rainfall makes a minimal contribution to the water balance (Power et al., 2015). The tremendous range of moisture states that occur in these seasonally dry regions ensures
that the study catchments "explore" a wide range of potential recession behaviors.

### 2.2 Overview of the methods varied across recession analyses

#### 2.2.1 Nomenclature and symbols used

To concisely describe the combinations of methods tested here, we first represent the four methodological choices with four binary (taking values of 0 or 1) variables:





1. Minimum recession length (M)

2. Peak selectivity (S)

3. Recession concavity (C)

4. Fitting method (L)

The extraction related variables (M, S, and C) are defined so that a value of 1 corresponds to a *more restrictive extraction method*; that is, the method choice filters out more recessions if its corresponding variable is 1 than if the variable is 0. For example, M = 1 corresponds to a minimum recession length of 10 days, which is more restrictive than a minimum recession length of 4 days (M = 0). Table 2 enumerates the 16 method combinations.

### 2.2.2    Defining the minimum allowable length of recession event (M)

Nearly all event-scale recession studies set a minimum duration for chosen recession periods. Reasons for this choice vary; authors cite the removal of noise from short events (Ye et al., 2014), the necessity of capturing late time flow processes (Chen and Krajewski, 2015), and data quality concerns related to sample size (Shaw, 2016). Event-scale recession analyses have typically chosen a minimum of 4 to 5 days of recession for daily data (e.g. Shaw and Riha, 2012; Biswal and Marani, 2010), although values as low as 12 hours have been used for high frequency data (McMillan et al., 2014).

To logically examine sensitivity to minimum recession length, the 'liberal' and 'restrictive' end member values should be chosen to be consistent with typical recession timescales of the study catchments. By fitting a linear recession model ($dQ/dt = -kQ$) to a representative collection of recessions from each catchment in our dataset, we find that median recession response timescales ($1/k$ [T]) range from about two to four days. To capture the important features of extracted recessions, while also varying the minimum recession length significantly with respect to typical response times (and also without choosing

values so restrictive as to limit the size of our sample sets), we extract restricted sets of recessions using a minimum length of ten days (M=1), and less restricted sets of recessions with a minimum length of four days (M=0).

### 2.2.3    Identifying potential recession starts (S)

Ideally, rainfall data would be used to identify periods of recession. However, high-quality precipitation records are often unavailable, and so the majority of event-scale recession analyses rely on flow data alone for recession identification. We

therefore only consider methods of recession analysis that can be applied to any daily streamflow record, with or without rainfall data. More stringent extraction methods that require rainfall data would be expected to reduce uncertainty in recession analysis, as extracted recession periods *with* rainfall data can reasonably be expected to be a subset of those extracted *without* rainfall data.

    Without rainfall data, recession starts are typically identified by locating days with discharge peaks; that is, times when

$dq/dt$ changes sign from positive to negative. However, some recession starts, while consistent with this definition, do not satisfy other important criteria for robust analysis and should be excluded. Rationales for exclusion might include discarding





minor peaks that are small relative to measurement error, or which have dynamics that would be expected to be unresolvable on daily timescales, although few authors give a strong justification for their choices in this regard. For example, Ye et al. (2014) discard peak flows less than the $10^{th}$ flow percentile to filter noise from small events. Mutzner et al. (2013) and Biswal and Marani (2010) choose only recession events where initial flow conditions are greater than mean annual flow in order to avoid

"minor events, which may not have significantly increased the average soil saturation, thus not triggering a significant response of the groundwater." Without identifying some justifiable tolerance for noise associated with small peaks, or defining what constitutes a significant groundwater response, it is difficult to objectively determine a peak threshold below which recessions should be excluded from analysis.

To test the effect of peak filtering decisions on recession analysis, we implement a peak selection procedure that is sensitive

to the "distinctness" of any given peak relative to the data around it (Yoder, 2009). Our scheme selects a peak if all of the following are true: (i) it is a local maximum; (ii) it is greater by some threshold amount ($X$) than the local minimum lying between it and the previously chosen peak; and (iii) discharge decays to a local minimum by the same threshold amount before the next greater local maximum is found. We define the threshold as $X = \text{range}(q)/d$, where $\text{range}(q) = \max(q)$ - $\min(q)$ is taken over the period of record. Here, $d$ is a tunable parameter that we set to be 50 for highly selective extraction (S=1; only

larger, more distinct peaks are analyzed) and set to 500 for less selective extraction (S=0, a broad range of peaks are analyzed).

In most studies, once a significant discharge peak has been identified, a recession start-time, which is often lagged from the discharge peak, is chosen. The most commonly cited rationale for this lagged recession start is to ensure the dominance of groundwater dynamics in the recession signal, rather than overland flow processes (e.g. Biswal and Marani, 2014; Patnaik et al., 2015). Most event-scale recession analyses lag recession starts by at least one day (Patnaik et al., 2015; Bart and Hope,

2014; Biswal and Marani, 2014), although it is not clear that such lagging is necessary to enable proper interpretation of event-scale dynamics (e.g. Harman et al., 2009). Fast flow processes, as well as slow, may also contribute to the hypothesized dynamics which generate power law recession behavior. For example, Harman et al. (2009) postulate that heterogeneous transport timescales alone give rise to power law recession dynamics, with no restriction on the "fastest allowable" timescale. Without *a priori* information that surface flow processes are a significant source of run off generation in a watershed, lagging

each recession start may be unnecessary. For example, no surface flow processes have been observed at the Elder Creek watershed in our collection of study watersheds; all runoff is generated by a highly responsive perched water table system (Salve et al., 2012). While adopting either approach – lagging the recession time or not – involves some risk, we seek and analyze distinct streamflow peaks without removing any days following the recession start.

### 2.2.4 Identifying the end of a recession event (C)

A number of criteria have been used to determine the end of a recession event. Without a reliable rainfall record, many event-scale analyses halt recession extraction upon the first day where flow does not decrease, that is, as soon as $dq/dt \geq 0$ (e.g. Mutzner et al., 2013). Vogel and Kroll (1996) define the recession end as the first day of increase in the 3-day moving average of streamflow. Shaw and Riha (2012) end the extracted recession two days before $dq/dt$ changes from negative to positive following a recession start. Some studies use the inflection point of the recession curve – the first day following a rainfall event



for which the hydrograph is concave down – to identify the *start* of the extracted recession (Singh and Stall, 1971; Wittenberg and Sivapalan, 1999). A similar concavity criterion, paired with the requirement of decreasing flow, could also be used to define the *end* of a recession event. Exploring every possible combination of the the above (and other) methods would lead to an intractably large number of methodological combinations. We therefore define two consensus strategies derived from the above criteria.

The first (C=0) considers a recession as any hydrograph segment with $dq/dt < 0$ following an identified peak. The second, more restrictive strategy (C=1) requires that the raw flow time series is strictly decreasing (again, $dq/dt < 0$) *and* classified as concave up. A recession day is classified as concave up if *either* the raw time series *or* a 3-day averaged time series is concave up; that is, if the second difference of either the raw flow time series or a smoothed flow time series is greater than or equal to zero. The inclusion of the criterion based on the three day moving average has the effect of including days with small "bumps" in concavity in the raw time series, while consideration of the raw time series ensures inclusion of days immediately after sharply peaked events, which are often classified as convex by the smoothed time series. This simple criteria could serve as an improvement to methods that only require $dq/dt < 0$, which could inadvertently extract highly convex recessions that are likely associated with continued rainfall.

### 2.2.5 Choosing a fitting procedure (L)

Fitting methods can be broken down into one of three categories: (i) linear regression or enveloping of a binned collection of $[\log(q), \log(-dq/dt)]$ points (e.g. Kirchner, 2009; Parlange et al., 2001); (ii) linear regression or enveloping of a raw collection of $[\log(q), \log(-dq/dt)]$ points (e.g. Brutsaert and Nieber, 1977; Biswal and Marani, 2010); or (iii) nonlinear regression (e.g. Wittenberg, 1994). Within these three general categories, a wide variety of specific regression techniques can be applied (e.g. Thomas et al., 2015; Zecharias and Brutsaert, 1988). Importantly, many of these approaches require a large number of data points and are thus unsuitable for event-scale methods.

For event scale recession fitting, the most popular method is to find a regression line through raw $[\log(q), \log(-dq/dt)]$ point data corresponding to each recession event. There is evidence, however, that nonlinear fitting methods produce more consistent values for recession parameter fits (Wittenberg, 1999; Chen and Krajewski, 2016). Moreover, nonlinear techniques have been used to successfully parameterize hydrologic models (Müller et al., 2014; Dralle et al., 2016), and to avoid numerical issues associated with computing the time derivative of a flow time series (Rupp and Selker, 2006b).

For the purposes of the present study, we again frame the problem in terms of the most fundamental methodological dichotomy between linear and nonlinear fitting. Linear fitting (L=1) is performed on the log-transformed values, $[\log(q), \log(-dq/dt)]$ (Brutsaert and Nieber, 1977). Nonlinear fitting (L=0) is performed on extracted, non-transformed recession segments.

### 2.3 Method combination comparisons

In general, only fitted exponents can be reliably compared between different recession events (e.g. Berghuijs et al., 2016; Sawaske and Freyberg, 2014). This is a consequence of a mathematical artifact that arises when fitting power laws to datasets with arbitrarily chosen scaling (Dralle et al., 2015). The issue can be avoided by setting the recession exponent to a fixed value





(e.g., the median (Biswal and Marani, 2010)), but this comes at the expense of biasing the fitted values of $a$ due to constraints on the exponent. Dralle et al. (2015) present a technique that removes the scaling artifact from the recession scale parameter without constraint on the recession exponent.

With this in mind, we choose three primary ***recession measures*** for comparison between recession events: the recession exponent ($b$), the scale-corrected (Dralle et al., 2015) recession scale parameter ($a$), and the recession time ($T_R$), defined by Stoelzle et al. (2013) as the amount of time required for flow levels to decline from the median flow to the tenth flow percentile. The measure $T_R$, which depends on both $a$ and $b$, belongs to a class of widely calculated recession timescales for the general, nonlinear form of Eq. (1) (e.g. Stoelzle et al., 2013; Westerberg and McMillan, 2015).

To see how methodological choices might impact the interpretation of $a$, $b$, and $T_R$, we organize our analysis around three primary questions:

1. *How do methodological choices impact the overall quality of individual recession fits?* – Fit quality is one measure of confidence in the estimated value for each recession measure. Testing event-scale recession theories that predict specific values for recession measures (e.g. Biswal and Marani, 2010) can be expected to be constrained by the degree of this confidence. This section looks to identify method combinations that consistently produce high quality fits, and thus high confidence in recession parameter estimates.

2. *Are a, b, and $T_R$ "characteristic" across various methodological choices?* – That is, do catchments rank in a similar order according to different statistical measures (in the present study, the median and inter-quartile range) of the populations of $a$, $b$, and $T_R$ across the sixteen method combinations (c.f. Stoelzle et al., 2013)? The results of comparative hydrologic studies (e.g. Bogaart et al., 2016), which rely on relative relationships between recession measures, can be expected to be affected by any methodological sensitivity demonstrated here.

3. *For each catchment, are the empirical frequency distributions of a, b, and $T_R$ statistically similar across method combinations, and, if not, what method choices have the greatest impact on recession parameter distributions?* – Beyond measures of central tendency of recession parameter estimates, event-scale theories suggest that higher order moments of recession parameter distributions should vary in systematic ways, depending on climate or catchment physiographic properties (Biswal and Nagesh, 2014; Harman et al., 2009). By addressing this question, we seek to identify the methodological choices which could most significantly impact testing of event-scale recession theories.

### 2.3.1 Testing the quality of recession fits

We report two measures of the overall quality of recession fits as a function of combinations of method choices. First we compute the mean average percent error (abbreviated as MAPE and denoted mathematically as $E_{\mathrm{MAP}}$) for each method combination, across all catchments. MAPE is computed as:

$$E_{\mathrm{MAP}} = \frac{1}{N} \sum_{i=1}^{N} \left| \frac{Q_i - \hat{Q}_i}{Q_i} \right|, \tag{2}$$



where $Q_i$ and $\hat{Q}_i$ are the observed and predicted flows on the $i^{th}$ day following the start of the recession event. (Note that comparing goodness of fit using $R^2$ is not appropriate, because one of our fitting methods is nonlinear (Kvalseth, 1985).) We also report, for each method, the percentage of all fits that yield "non-physical" estimates for the recession parameters, which we define as $b < 0$. In all subsequent analyses, the recession parameters are filtered so that $b \geq 0$ ($b < 0$ occurs for less than 3%

of all recession events).

### 2.3.2 Ranking catchments by recession characteristics

While Stoelzle et al. (2013) perform lumped recession analysis and obtain single recession parameter values for each catchment and method combination, our event-scale analysis yields distributions for $b$, $a$, and $T_R$. We therefore report measures of central tendency *and* variability for the computed recession variables, $b$, $a$, and $T_R$. Following Stoelzle et al. (2013), we compute

Spearman rank correlation coefficients by ranking catchments between method combination pairs based on the following measures (recession characteristics): median($a$), median($b$), median($T_R$), IQR($a$), IQR($b$), IQR($T_R$), where IQR is the interquartile range.

Even if the absolute magnitudes of the values of $a$, $b$, and $T_R$ vary between the method combinations, these rank tests will determine whether catchments rank in the same order by the recession characteristic for all methods. Determining the

consistency of such ranked comparisons has implications for efforts to develop effective metrics for catchment classification, where relative differences in recession characteristics have been used to compare or categorize catchments (e.g Bogaart et al., 2016; Mutzner et al., 2013; Guzmán et al., 2015)

### 2.3.3 Comparing distributions of $a$, $b$, and $T_R$ across method combinations

While shifts in the Spearman rank correlation between method combinations allow a comparative analysis of the effects of

method choice, they do not provide information about variations in the specific values of the recession parameters obtained by each method. To address the specific values of the recession parameters, which is important for testing theories that make such specific predictions (Biswal and Marani, 2010; Brutsaert, 1994), we therefore also explore the empirical frequency distributions of parameter populations estimated with each methodological combination.

We first illustrate general patterns of $a$, $b$, and $T_R$, across all method combinations with Tukey box plots for a single rep-

resentative catchment – the Elder Creek watershed, a tributary of the Eel River in Northern California. These plots provide visual representation of the *observed* difference between the character of recession measure distributions for different method combinations. However, they do not represent the absolute effect of changing individual method choices. This is because more restrictive extraction procedures produce populations of recessions that are a subset of the populations generated by less restrictive extraction measures. For example, fixing all other method choices, recessions extracted with a minimum length of 10

days must be a subset of recessions extracted with a minimum length of 4 days. This "dilutes" the true effect of the shift in choice of minimum recession length on the recession measures derived from the two resulting populations.

One way to isolate the absolute effect of a given method choice is to compare recessions that are *shared* between the restrictive choice and non-restrictive choice, to those that are *unshared* between the restrictive and non-restrictive choices.





This procedure is illustrated for the minimum length choice in Fig. 1. Here, the raw streamflow data (Fig. 1a) is subjected to extraction procedures with a minimum length of 4 days (Fig. 1b) and with a minimum length of 10 days (Fig. 1c). All other method choices are fixed. Clearly the 10 day minimum length recessions are a subset of the 4 day minimum length. Two distinct groups can then be formed: a set of recessions *shared* between the two extractions (Fig. 1c), and a set of *unshared* recessions

(Fig. 1d; those extracted by the minimum 4 day extraction, but not the 10 day extraction). Differences between these disjoint "shared" and "unshared" sets of recessions embody the absolute effect of an individual method choice on a recession measure. Recession measures between the two groups should be comparable if the particular recession measure is not sensitive to the method choice.

We compare shared and unshared recession measure distributions in two ways. First, for a high level overview, we show

Tukey box plots of shared vs. unshared distributions of the recession exponent ($b$) for a single catchment (the Elder Creek watershed) for each of the recession extraction choices (M, S, and C). We also compare populations between linear and nonlinear fitting, though we note that this is not a "shared" vs. "unshared" comparison.

We then use a two-sided Mann-Whitney U Test (Mann and Whitney, 1947) to compare shared vs. unshared distributions for each recession measure across all method choices and all catchments. The null hypothesis for this non-parametric test is

that the shared and unshared distributions are sampled from the same population. For a given catchment and for each method choice, we compute p-values for the Mann-Whitney U Test by comparing shared to unshared distributions for each of the eight combinations of the other method choices. If the test rejects the null hypothesis, then we conclude that the method choice significantly changes the distribution of the recession measure.

For each method choice and catchment, we then compute the fraction of tests (eight total tests per method choice per

catchment; shared vs. unshared for all 8 combinations of the other method choices) which returned statistically different shared and unshared distributions. We use this fraction as an indicator of the sensitivity of a recession measure to a given method choice. We perform this procedure for all recession measures, $a$, $b$, and $T_R$. Since each measure requires 512 total comparisons (16 catchments $\times$ 8 tests $\times$ 4 method choices), we apply a Bonferonni correction for the critical p-value of each test, which is required when a statistical test is applied many times for multiple comparisons (Abdi, 2007). For an overall level of significance

of $\alpha = 0.05$, the correction requires a critical p-value for each test set to $p = \alpha/512$.

## 3 Results

Catchments were ranked by the values of 6 recession characteristics – median($a$), median($b$), median($T_R$), IQR($a$), IQR($b$), and IQR($T_R$) – for all pairs of method combinations. The collection of corresponding Spearman rank correlations are presented as box plots in Fig. 2. The rank correlation can take a value between -1 and 1, where a rank correlation of 1 indicates that two

methods produce identical rankings and a rank correlation of -1 indicates that two methods produce exactly opposite rankings. We performed a thorough investigation of the rank correlations between different method combinations across all 16 study catchments but found few patterns related to individual method choices. Therefore, we present aggregated box-plots of the Spearman rank correlation for each of the recession characteristics. Overall, none of the rank correlations were negative,



suggesting that, at worst, no method combination predicts a characteristic ranking that is inverted relative to another method combination. The most "characteristic" measure, in the sense that its ability to rank catchments is least sensitive to the method choice, is median($a$).

The box-plots in Fig. 3 are generated using computed MAPE values for all fits from each combination of method choices.
The figure provides a rough measure of the sensitivity of fit quality to each individual methodological choice. The patterns in Fig. 3 hint at a hierarchy of the importance of method choices in terms of their impact on the "quality" of extracted recessions and their corresponding power law fits. Specifically, the concavity (C) and linearity (L) method choices roughly subdivide the results into three groups: The worst fits observed were those performed without the concavity requirement and with linear regression (shown as combinations that end in 01 in Fig. 3). Fits that use concave recessions or nonlinear fitting, but not both,
are of intermediate quality (combinations that end in 00 or 11). The best fits by a large margin are those that combined the concavity requirement with nonlinear regression (combinations that end in 10).

Figure 4 contains box plots for all three recession measures and for all method combinations for the Elder Creek watershed. The overall patterns observed here are comparable to those of the other 15 watersheds. While the scale-correction procedure for $a$ has only been applied in one previous study (Dralle et al., 2015), the median values of scale-corrected $a$ are consistent
with inverse recession timescales (commonly referred to as the 'recession constant') extracted from linear reservoir models (e.g. Sánchez-Murillo et al., 2015; Botter et al., 2013). The observed median values of $b$ and $T_R$ are also consistent with those typically found in lumped recession analyses (e.g. Tague and Grant, 2004; Palmroth et al., 2010; Szilagyi et al., 2007; Wang, 2011; Stoelzle et al., 2013; McMillan et al., 2014). Variability in the recession measures can be significant. The recession exponent $b$ regularly falls between $b = 1$ and $b = 2.5$, while the inter-quartile range for $a$ and $T_R$ span upwards of an order of
magnitude. This degree of variability in $a$, while large, is comparable to event-scale recession studies that impose a fixed value on the recession exponent (e.g. Shaw and Riha, 2012).

Figure 5 presents box plots for the shared and unshared distributions of the recession exponent for the Elder Creek watershed. Each subplot in Fig. 5 corresponds to one of the four method choices (M, S, C, or L). The light green boxes represent the distribution of the recession exponent for shared recessions, while the dark green boxes represent the distribution of the
recession exponent extracted by only the less restrictive procedure. The horizontal axes in each subplot show the eight possible combinations of the other method choices, showing how these shared and unshared distributions vary for different combinations of the other method variables. Significant differences between the shared and unshared distributions in Fig. 5 indicate that the recession measure is sensitive to modulation of the method choice represented by the particular subplot.

The interpretation of "shared versus unshared" is different for each method choice. For the minimum recession length,
"not shared" means 4-9 day recessions; and "shared" means recessions 10 days or longer. For peak filtering selectivity, "not shared" means the many small flow peaks not extracted by the more selective procedure; "shared" means large flow peaks. For concavity, "not shared" means recessions that are not distinctly concave (which, presumably, are the recession periods with residual precipitation); "shared" means drier recessions, or recession periods that are less likely to have significant residual precipitation.





Results of the Mann-Whitney U tests between shared and unshared distributions for each recession measure, for all method choices, and for all catchments, are presented in Fig. 6. The Elder Creek (catchment number 11475560) recession exponent distributions in Fig. 5 correspond to the recession exponent subplot of Fig. 6. In agreement with Fig. 5, the recession exponent is most significantly affected by the choice to extract only concave recessions. The strong dependence on concavity demonstrated

in Fig. 5 manifests in Fig. 6 as the very dark rectangle in the concavity column of the recession exponent parameter for catchment 11475560. This indicates that all Mann-Whitney U tests detected a significant difference between the eight shared and unshared distributions for concavity.

## 4 Discussion

### 4.1 Recession fit quality

The finding that concavity and linearity play primary roles in determining the quality of recession fits is notable in light of the fact that minimum recession length and minimum recession peak size are more commonly emphasized as the most important methodological choices made during event-scale recession analysis (e.g. Biswal and Marani, 2010; Patnaik et al., 2015; Mutzner et al., 2013). Evidence here suggests concavity requirements and nonlinear fitting greatly improve the quality of event-scale recession analyses and that these improvements are additive when we impose these methodological choices

together. In fact, the often-used definition of flow recession, that the flow derivative is negative, could be misleading; the simple dynamical system model developed by Kirchner (2009) predicts that streamflow can decrease during precipitation events. The use of improved, flow-derived recession extraction methods, such as the concavity requirement, could reduce the frequency of "false" recession extraction, increasing the quality of recession measure estimates.

Beyond the tendency to produce lower quality fits, the linear fitting procedures applied in the majority of recession studies

have other well-documented drawbacks. Linear regression on log-transformed flow values disproportionately weights errors for smaller model values, creating a risk of bias in the fit (Miller, 1984; Pattyn and Van Huele, 1998). Linear fitting also requires computation of the flow derivative, which introduces a number of documented numerical and data quality challenges (Rupp and Selker, 2006b). The various differencing schemes that can be implemented to obtain the flow derivative (e.g. Thomas et al., 2015) add another potential source of method dependent bias in the fitting scheme. There are downsides, however,

associated with nonlinear fitting (Motulsky and Ransnas, 1987). Fit bias may be introduced by the optimization algorithm used, or the necessity of specifying an initial condition for the nonlinear fitting procedure. Choice of initial values can be relatively clear for recession measures like $b$ that can be expected to have tightly constrained values, but for other more variable recession measures, such as $a$, this choice could also be opaque, and differing initial conditions could lead to differing recession parameter estimations.



### 4.2 Recession measures are characteristic

We find that the medians and IQRs of $a$, $b$, and $T_R$ are all fairly characteristic, though to varying degrees; median($a$) is more characteristic than median($b$) or median($T_R$), and each IQR is less characteristic than its corresponding median. One might expect that median($a$) is highly characteristic because it spans many orders of magnitude, while other parameters are more

tightly constrained. However, the derived measure $T_R$ also has a wide range and does not display the same level of rank stability as $a$; see Fig. 4 for approximate ranges.

The relatively stable ranking of catchments by recession measures has potential implications for testing event-scale recession theory. Recent work by Harman et al. (2009) hypothesizes that $b$ can be interpreted as a measure of the diversity of water transport timescales throughout the various parts of the catchment. In this framework, measures of variability of $b$ could be

interpreted as representative of the "realizable" range of catchment states, with respect to the relative dominance of various water transit times in the catchment. Strongly characteristic measures of $b$ suggest the potential to use the recession exponent to develop relative measures of catchment complexity, if the Harman et al. (2009) theory applies to catchment populations.

Results also provide support for application of the recession scale parameter scale-correction procedure presented by Dralle et al. (2015). Medians of the scale-corrected recession scale parameters rank catchments more consistently than all other

recession characteristics. Moreover, the fact that $a$ has units of inverse time suggests it can be interpreted physically in a manner similar to more commonly computed response timescales, such as $T_R$ (e.g. Stoelzle et al., 2013; Westerberg and McMillan, 2015). In fact, the median and IQR of $T_R$ are the least consistent catchment ranking characteristics. Considering that $T_R$ is a measure derived from both $a$ and $b$, it has likely inherited catchment ranking uncertainty from both these parameters. Numerous derived recession measures have been used for comparative catchment analysis (Sawaske and Freyberg, 2014; Berghuijs et al.,

2016; Stoelzle et al., 2013), and the findings here suggest a trade-off; the development of more complex derived measures comes at the risk of compounding uncertainty.

### 4.3 Comparing distributions of recession measures

The repeating "saw-tooth" pattern for $b$ seen in Fig. 4 indicates that concavity and linearity play important roles in shifting the distributions of the recession exponent. If other methodological choices are fixed, linear fitting and concavity both produce

noticeably higher values for the recession exponent. Without the concave requirement, the "decreasing only" extraction procedures will produce lower values due to increased convexity. Table 3 supports this conclusion; the concavity requirement greatly decreases the number of "non-physical" ($b < 0$) extracted recessions. The upward shift for linear fitting may be the result of "over-weighting" of errors in the tail end of the recession, where deviations from linearity in the curve defined by the collection of $[log\,(q)\,,\log\,(-dq/dt)]$ point pairs are consistently observed to exhibit a steeper slope. This supports the use of non-linear

regression techniques as a means to avoid biases inherent in log-transformed power law fits.

The pattern of shorter whiskers from left to right in Fig. 4 shows that the variability of the recession measures decreases as extraction procedures become more restrictive. For a minimum recession length of 10 days and highly selective peak filtering (M=1, S=1), this decrease in variability is likely due to the fact that the collection of extracted recessions becomes less "diverse"




as the extraction method becomes more restrictive, as suggested by Stoelzle et al. (2013). As compared to minimum length and peak selectivity, which had a minimal impact on fit quality (see Fig. 3), the larger variability for non-concave data paired with nonlinear fitting is due, at least in part, to more noise from persistent rainfall during the recession. This suggests again that peak size and recession length data quality concerns cited by some authors (e.g. Ye et al., 2014; Shaw, 2016) could be

augmented to consider fitting methods and the "quality" of the shape of extracted recessions.

Patterns displayed in Fig. 5 are largely similar to distributions of $b$ in other watersheds. Whiskers are longer for "shared" distributions for minimum length and selectivity. This could be because there are typically fewer large storms and long recessions than there are small storms and short recessions, or because very large storms and very long recessions represent the asymptotic behavior of the catchment response associated with more extreme states. The length of concavity whiskers are comparable

between shared and unshared distributions, although concavity again emerges as the most important choice for determining the absolute magnitude of $b$. There is a clear separation between shared and unshared distributions of $b$ for concavity.

While certain method choices seem to play important role in determining quality of fit, Fig. 6 demonstrates that other choices could play a more important role in determining realized values of $a$ and $b$. This finding makes it difficult to determine the "best" method combination. Whereas concavity and linearity were the dominant drivers of goodness of fit, it is selectivity and

concavity that exert the strongest control over the distribution of $b$. Minimum recession length seems to exert the strongest control over the distribution of the recession scale parameter ($a$). Along with the inconsistencies in controls on each recession measure, we also note that some recession measures are uniformly sensitive to a given method for all catchments (e.g. concavity strongly affects $b$ for all catchments), while others seem to vary between catchments. For example, linearity exerts a strong control on the distribution of $b$ for catchment 11468500 (Noyo River), but apparently makes very little difference for catchment

11143000 (Big Sur River).

## 4.4 Consequences for event-scale recession theory

The "revival" of power law recession analysis at the event-scale can be attributed primarily to two new (and distinct) theories concerning catchment function, both of which predict that recessions should take a power law functional form, and that the recession parameters $a$ and $b$ should vary between events. Clark et al. (2009) and Harman et al. (2009) theorize that power

law parameters provide information primarily about the partitioning and distribution of flow residence timescales within the catchment. Logical extensions of the theory suggest that measures of variability in the recession exponent could provide information on heterogeneity in catchment transport timescales. Contrasting theory by Biswal and Marani (2010) suggests that the recession scale parameter can be uniquely mapped to the rate of wetted channel contraction during the recession phase. In later work, Biswal and Nagesh (2014) hypothesize that spatial heterogeneity in rainfall could also introduce variation in the

recession exponent.

The theory of Harman et al. (2009) and Clark et al. (2009) has some support from abstract, multi-reservoir models, but it is unclear precisely how such models correspond to the physical architecture of a catchment. A key prediction made by Harman et al. (2009) – that the recession exponent should increase with catchment size – receives mixed support from empirical studies (e.g. Stoelzle et al., 2013). In comparison, the wetted channel dynamics underlying the theory of Biswal and Marani (2010) are



highly observable, although some empirical evidence has been presented that contradicts important assumptions of the theory (Shaw, 2016; Whiting and Godsey, 2016).

Overall, few studies have attempted to tease apart the convergent predictions of power law recession theories. Some works informed by Biswal and Marani (2010) demonstrate a relationship between measures of antecedent catchment wetness and

the power law scale parameter (e.g. Bart and Hope, 2014; Biswal and Nagesh, 2014; Patnaik et al., 2015), although explicit connections to wetted channel network expansion and contraction still require elucidation (Ghosh et al., 2016). Whatever its physical basis, we observe similar correlations between measures of antecedent wetness and the scale-corrected recession scale parameter. Figure 7 plots the recession scale parameter versus a measure of antecedent wetness for the Elder Creek catchment for three methodological combinations. The antecedent wetness measure $(W)$ is computed as a weighted sum of streamflow

prior to each recession event:

$$W = \sum_{i=1}^{60} 0.95^i Q_i, \tag{3}$$

where $i$ is the number of days prior to the start of the recession event. The weighting coefficient, $0.95^i$, is included to discount the effect of less recent events on the catchment wetness state. All plots demonstrate a decreasing, log-log linear relationship between the antecedent wetness measure and the recession scale parameter. The first and second plots are less restrictive with

respect to recession length and peak size; the first plot extracts concave recessions and uses nonlinear fitting, while the second plot extracts decreasing recessions with linear fitting. The third plot uses a highly selective extraction procedure and fits the recession model with nonlinear regression. Despite moderate sensitivity to concavity and linearity in the Elder Creek data (catchment 11475560) displayed in Fig. 6, the first two fits are not statistically different as shown by 95% confidence intervals for the fitted slopes. The slope on the third plot differs significantly from the first two, likely due to the fact that the population

of recessions originates from a highly selective extraction procedure. This highlights the potential for recession extraction bias in parameter populations; without good cause to discard smaller or shorter recessions, such choices can lead to quantitatively different interpretations of recession parameter values.

## 5 Conclusions

This study quantified the sensitivity of the power law streamflow recession parameters $a$ and $b$ to four common methodological

choices made during recession extraction and fitting. While rankings of study catchments in terms of the descriptive statistics of $a$ and $b$ were relatively insensitive to the methods used, individual method choices did significantly impact observed parameter distributions, though each parameter had a distinct sensitivity profile. These results highlight the importance of accounting for methodological uncertainty when performing event-scale recession analysis.





## 6   Data availability

All streamflow data used for this study can be found on the website for United States Geological Survey (http://waterdata.usgs.gov/nwis).

*Author contributions.*

5   *Acknowledgements.*  The authors would like to thank Davit Khachatryan and Laurel Larsen for helpful conversations concerning some aspects of the statistical and sensitivity analyses presented here.




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





**Table 1.** Study catchments

| USGS gage number | Catchment name | Catchment area [km$^2$] | Number of years of data |
|---|---|---|---|
| 11143000 | Big Sur River, Big Sur, CA | 120.4 | 65 |
| 11451100 | North Fork Cache Creek, Clearlake Oaks, CA | 155.9 | 44 |
| 11463170 | Big Sulphur Creek, Cloverdale, CA | 33.9 | 35 |
| 11468000 | Navarro River, Navarro, CA | 784.8 | 65 |
| 11468500 | Noyo River, Fort Bragg, CA | 274.5 | 64 |
| 11472200 | Outlet Creek, Longvale, CA | 416 | 37 |
| 11473900 | Middle Fork Eel River, Dos Rios, CA | 1929.5 | 50 |
| 11475000 | Eel River, Fort Seward, CA | 5457.1 | 60 |
| 11475560 | Elder Creek, Branscomb, CA | 16.8 | 48 |
| 11476500 | South Fork Eel River, Miranda, CA | 1390.8 | 75 |
| 11476600 | Bull Creek, Weott, CA | 72.8 | 55 |
| 11481000 | Mad River, Arcata, CA | 1256.1 | 65 |
| 11481200 | Little River, Trinidad, CA | 104.9 | 60 |
| 11482500 | Redwood Creek, Orick, CA | 717.4 | 62 |
| 14307620 | Siuslaw River, Mapleton, CA | 1522.9 | 48 |
| 14325000 | Coquille River, Powers, OR | 437.7 | 99 |





**Table 2.** Enumeration of the methodological choices associated with the sixteen method combinations considered here.

| Method combination | Minimum recession length (M) $M = 1 \implies$ min len = 10 days $M = 0 \implies$ min len = 4 days | Peak selectivity (S) $S = 1 \implies d = 50$ $S = 0 \implies d = 500$ | Recession concavity (C) $C = 1 \implies$ concave and decreasing $C = 0 \implies$ decreasing | Fitting method (L) $L = 1 \implies$ log-log linear fitting $L = 0 \implies$ non-linear fitting |
|---|---|---|---|---|
| 0 | 0 | 0 | 0 | 0 |
| 1 | 0 | 0 | 0 | 1 |
| 2 | 0 | 0 | 1 | 0 |
| 3 | 0 | 0 | 1 | 1 |
| 4 | 0 | 1 | 0 | 0 |
| 5 | 0 | 1 | 0 | 1 |
| 6 | 0 | 1 | 1 | 0 |
| 7 | 0 | 1 | 1 | 1 |
| 8 | 1 | 0 | 0 | 0 |
| 9 | 1 | 0 | 0 | 1 |
| 10 | 1 | 0 | 1 | 0 |
| 11 | 1 | 0 | 1 | 1 |
| 12 | 1 | 1 | 0 | 0 |
| 13 | 1 | 1 | 0 | 1 |
| 14 | 1 | 1 | 1 | 0 |
| 15 | 1 | 1 | 1 | 1 |



**Table 3.** Fraction of recessions with non-physical recession exponent ($b < 0$) for each method combination.

| Method combination (MSCL) | Fraction of fits with $b < 0$ |
|---|---|
| 0 (0000) | 0.114 |
| 1 (0001) | 0.035 |
| 2 (0010) | 0.070 |
| 3 (0011) | 0.005 |
| 4 (0100) | 0.097 |
| 5 (0101) | 0.024 |
| 6 (0110) | 0.059 |
| 7 (0111) | 0.003 |
| 8 (1000) | 0.032 |
| 9 (1001) | 0.002 |
| 10 (1010) | 0.013 |
| 11 (1011) | 0.0 |
| 12 (1100) | 0.016 |
| 13 (1101) | 0.001 |
| 14 (1110) | 0.004 |
| 15 (1111) | 0.0 |


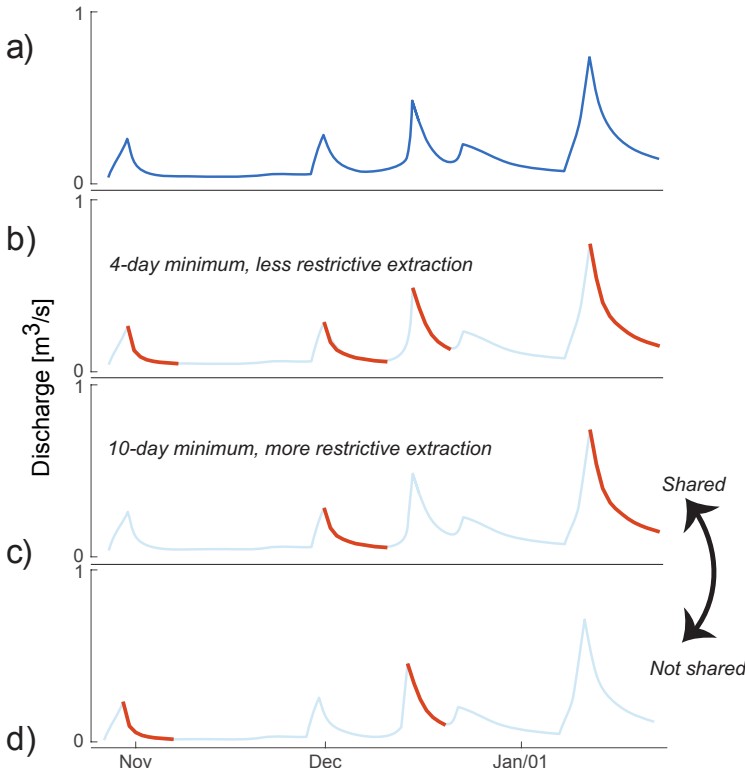

**Figure 1.** Example recession extraction from the hydrograph (a) using a less restrictive method M=0 (b) and a more restrictive method M=1 (c). The recessions identified by the more restrictive method will be "shared" by the two methods, in the sense that they will by definition also be identified by the less restrictive method. Recessions identified by only the less restrictive method (d) are classified as "not shared".





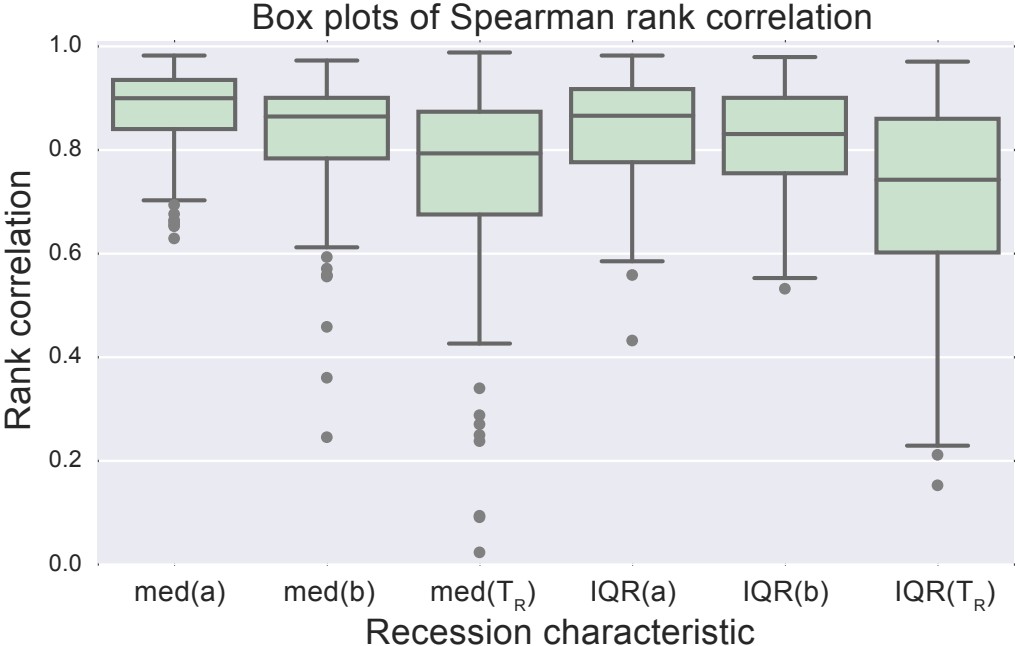

**Figure 2.** Box and whisker plots of Spearman rank correlations for all six descriptive measures of the distributions of $a$, $b$, and $T_R$. Per characteristic, there are $15 \times 16 = 240$ unique pairwise comparisons between method combinations.





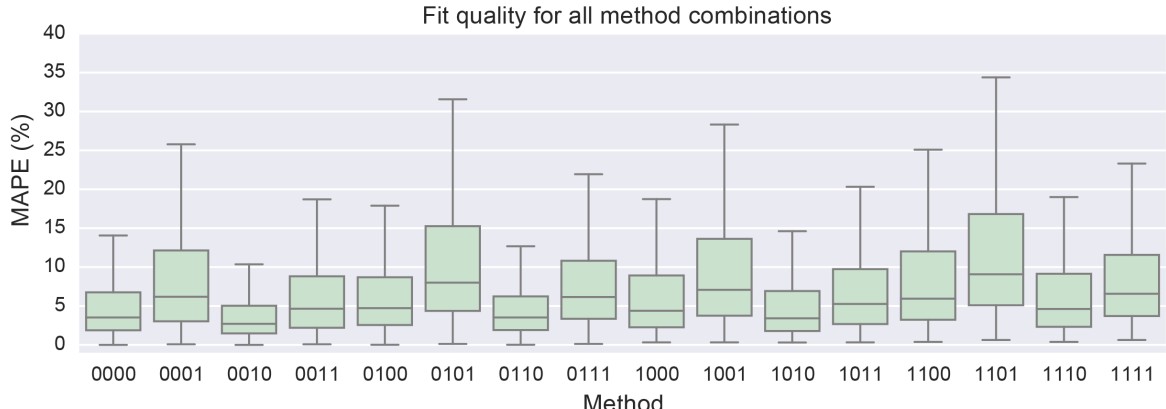

**Figure 3.** Mean absolute percentage error (MAPE) for all method combinations, lumped across catchments.





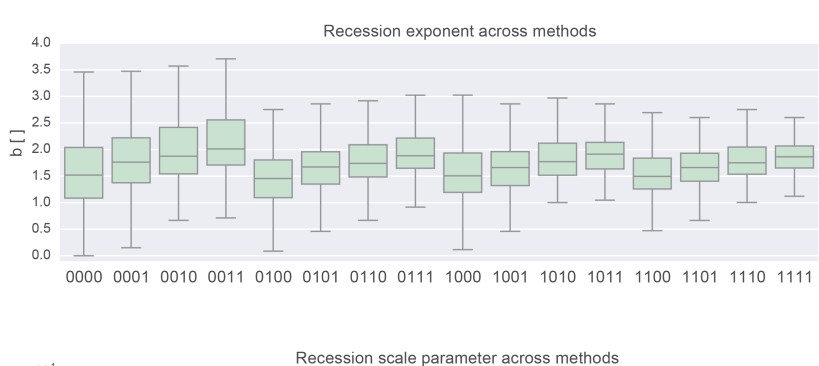

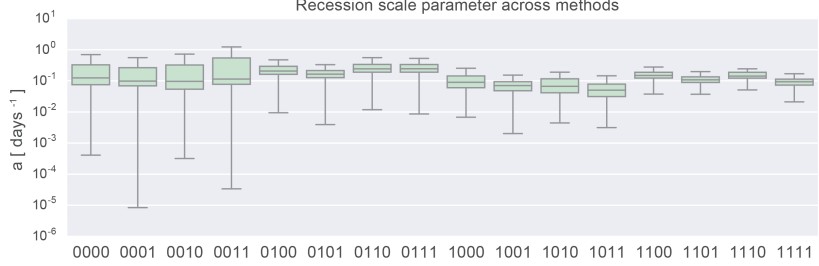

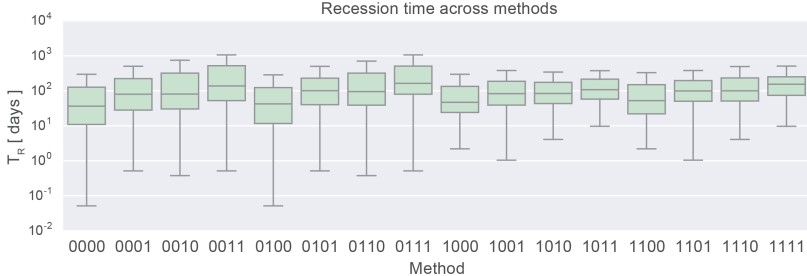

**Figure 4.** Box plots for $a$, $b$, and $T_R$ across all method combinations for Elder Creek watershed.





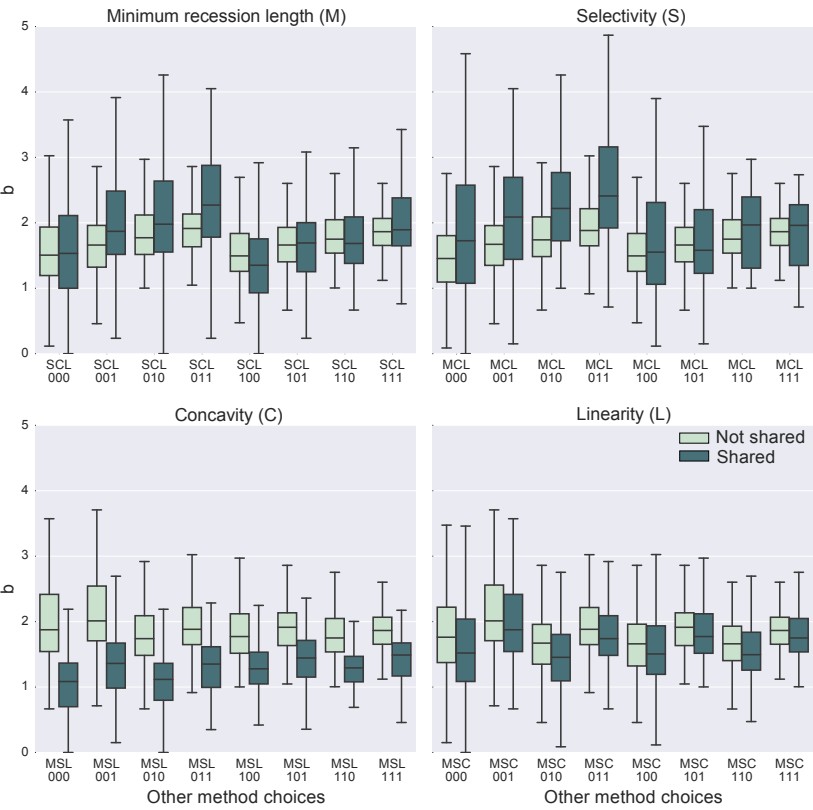

**Figure 5.** Box plots comparing shared vs. unshared distributions for the recession exponent for Elder Creek. Each sub-plot corresponds to a particular method choice; the shared boxes are generated with the $b$ values from the recessions shared between the 0 and 1 values of the subplot method choice. The unshared boxes are those values of $b$ from the recessions extracted by only the less restrictive value of the subplot method choice. The independent axis shows the values for the method choices other than the subplot method choice. For the linear versus nonlinear comparison, the distributions are by definition all shared, as the L method choice does not affect the extraction procedure; here the comparison is simply between population of nonlinear fits (0) and linear fits (1).





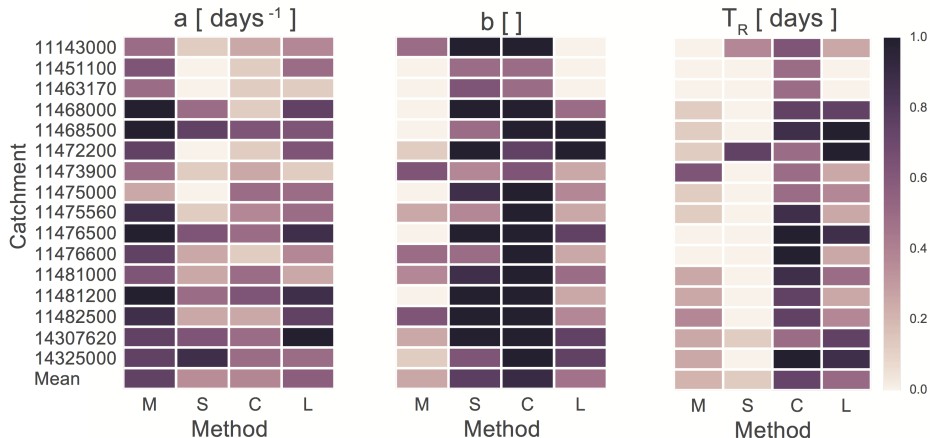

**Figure 6.** Results of Mann-Whitney U test sensitivity analysis. Each row represents one of the 16 study catchments, each subplot one of the three recession measures $a$, $b$, or $T_R$, and each subplot column one of the four methodological choices (MSCL). Each cell is colored by the fraction of statistically different pairs of shared versus unshared distributions for the particular recession measure conditioned by the two values of the corresponding method choice. A cell shading of 1 (dark purple) means all eight pairs of shared and unshared distributions were determined to be statistically different, indicating that the particular recession measure is highly sensitive to the corresponding method choice.





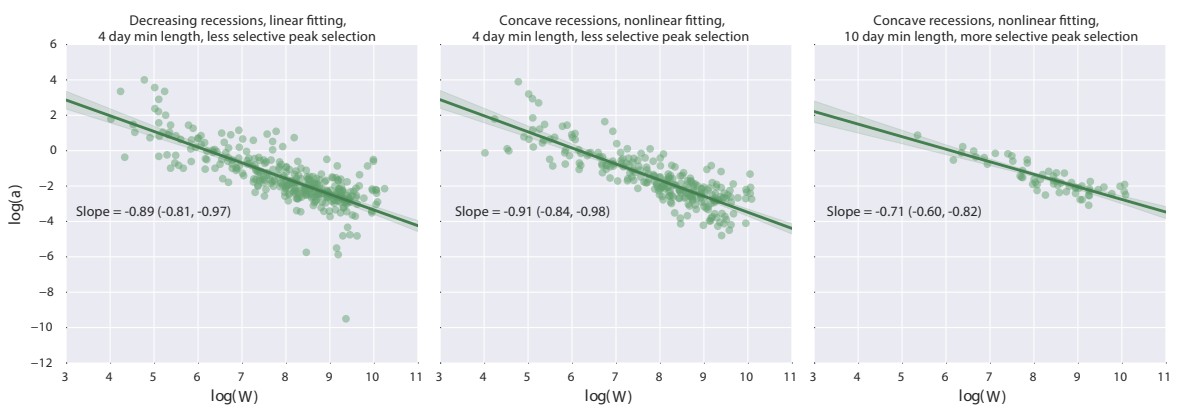

**Figure 7.** The recession scale parameter plotted against antecedent catchment wetness for three method combinations, together with a linear fit for each point cloud, and a 95% confidence interval for each fitted slope.