# Peer review of "Event-scale power law recession analysis: Quantifying methodological uncertainty"

_Hydrology and Earth System Sciences, 2016_

## Referee Comment (RC1) · Anonymous Referee #1 · 12 Sep 2016

Review of Event-scale power law recession analysis: Quantifying methodological uncertainty by Dralle et al.

The authors state that, while there is increasing attention for single streamflow recession characterization, it is unknown what the dependence of estimated model parameters on methodological choices is. To resolve this problem, they use daily streamflow data from 16 catchments located in California and Oregon and investigate how commonly used streamflow recession definitions influence the parameters of a power-law recession model that describes the recession. The methodological choices include: - the start of a recession - the end of a recession - the minimum length of a recession - the method of power law model fitting Results indicate that these choices can impact parameter value estimates, whereby the recession parameter distributions are method-dependent, but a particular method affects a given parameter in similar ways

across most catchments.

The article is generally well written and addresses a relevant topic (i.e. suitable for HESS). While I am generally positive about this article I think a few things need to be clarified before I can recommend publication in HESS.

- The article lacks information of the hydrology of the catchments beyond its approximate location, size, and that they are in a Mediterranean climate. Some more hydrological context will help to better understand the (lack of) transferability (to other catchments) of your results. For example I can imagine that these methodological choices may have a different influence for an arid headwater catchment vs a wet large catchments etc. etc. When you provide some additional information about the catchments, I also expect that you put your results into context of the range of catchments that you cover, i.e. how generalizable do you think your results are, or are they only representative for this small subset of catchments?

- The goal of event-scale recession analysis is to interpret variations in catchment response to rainfall as a function of the properties of rainfall events or the catchment state. Because your methodological choice will affect what information you obtain from the recessions (e.g. do you include the recession just after a peak, or do you wait a few days can affect if you include the information on quicker flow processes) I expect that such choices are explicitly discussed in context of single recession analysis (and thereby you differentiate your work better from e.g. Stoelzle et al, who have done something similar as you present but then for a cloud of points).

- Given the (approximate) location of your catchments I suspect that snow may play a role in streamflow generation. Do you need to account for this in determining the hydrograph recession periods? If yes; please apply a method that takes the role of snow into account. If no: make clear why snow is irrelevant for your study.

Technical/minor comments:

- To my understanding Ye et al. (2014) do not use individual recession events but they use the lumped version instead.

Please check your reference list. There is missing information (e.g. page numbers) and wrong information (wrong journal). This list below is therefore not complete.

- Berghuijs, W., Hartmann, A., and Woods, R.: Streamflow sensitivity to water storage changes across Europe, Water Resources Research, doi:10.1002/ 2016GL067927, 2016 is published in GRL, not WRR.

- Stoelzle, M., Stahl, K., and Weiler, M.: Are streamflow recession characteristics really characteristic?, Hydrology and Earth System . . . , 2013.: "Misses information"

- Whiting, J. A., and Godsey, S. E. (2016): add "30: 2305–2316. doi: 10.1002/hyp.10790." to reference

---

## Referee Comment (RC2) · M. Stoelzle (Referee) · 30 Sep 2016

The authors present a well-written study of an event-scaled recession analysis method. The paper is in the scope of HESS and interesting for readers that focus on low flows, recession behavior, drought and hydrograph analysis. In the paper recession analysis is subdivided into recession extraction and the fitting procedure. Then two "configuration sets" for each of four procedures are used to evaluate what aspect of recession analysis influences the projected recession behavior most. The method description is complete enough to reproduce the approach. However, the paper has it weaknesses in structure, mixture of content in different sections and in the visualization of the results. I have doubts if the choice of graph types (only boxplots) to illustrate the work is the best practice. I suggest consideration for publication after moderate revisions. The paper fails - until now - to put the developed method into a valuable perspective

for other researches and hydrological analyses.

Major comments

- The introduction is rather short, but leads directly to the mentioned research gap. However, please consider adding some more details why exactly the four mentioned methodological choices are most important to conduct a reasonable event-scale recession analysis? Also some more information about the choice of a "good" catchment set (with specific characteristics) is missing here, since one could argue that length, beginning and end (definitions 1-3) of recession events depend on the specific streamflow regime of a catchment. Would the results change if one compare rainfall- to snowfall-dominated catchments and so on?

- I would assume that with the description of the climate region (P3L22) a pronounced streamflow regime with low inter-annual streamflow variability could be expected. In the perspective of an improved "event-scale" analysis I wonder why these regime type would be very beneficial for such an analysis? Perhaps it would be better to use more erratic streamflow regimes since they will "produce" more distinguishable recession events with different features? Please comment on that in section 1 or 2. An overview of the catchments - or even better a simple graph showing the regimes or the variability of streamflow would be useful for the reader.

- I really like the idea of the binary nomenclature. However, for the reader it would be easier to present a direct link to the for variables: 1111 could be "MSCL", 0000 could be "mscl", 0101 could be "mScL", so using lower case for e.g. the more restrictive feature an upper case for the more versatile feature in each of the four classes M, S, C and L. I know this would be a little bit of work, but it is worthwhile to consider this as easier, more direct "coding".

- The peak filtering approach is very interesting. The only reference the authors gave here is a Math Works webpage. As a good understanding of the peak filtering procedure is needed to judge the results and discussion later on, it would be helpful to

have a small sketch/graph about this procedure as long as no other literature on that approach exists. The readership will also benefit if 2.2.4 (recession end) is somehow illustrated.

- The research questions are placed into the Method section (P7L11-L26). This is not common practice. I recognized that these questions are derived from the developed method catalogue above, but for the sake of clarity and a good paper structure these questions should be firstly outlined at the end of the Introduction. There the reader is anyhow a little bit disoriented without more guidance, what questions this paper wants to answer later on.

- The authors stated that the median and IQR is appropriate to evaluate the ranking of the catchments according to the "recession behavior". Why are these statistical metrics used? Is that the best choice to evaluate and answer question 2? (P7L16-20). Later on (2.3.2) the Spearman rank correlation is introduced. Please streamline these method explanations, they are mixed up a little bit.

- The methods section is well written and the reader can follow the idea of analysis. The authors could, however, consider to present some kind of flow chart that explains the 16 method combinations, the shared and unshared analysis, the MWU test, the 512 comparisons and so and so on. As this paper present novel and interesting methods for recession analysis, it seems to be important to illustrate an overview of the complete method/approach.

- I found very often a mixture of results and discussion (e.g. P10L12-21), please put some effort into the revised version to clarify the content of different sections (method, results, discussion).

- The streamflow data length varied greatly between the catchments (35-99 years of data). It would be better to show the periods of record for each catchment. If data is not collected more or less during comparable periods, the authors should comment on the potential effect of different data periods and different data length (e.g. recession

analysis in very long time series can be blurred by changes of recession behavior over time or recession behavior in different decades could be a caused for different parameterization later on).

- In summary the tables and graphs are rather poor -or at least a little bit uninspired -illustrations of a very interesting and comprehensive analysis.

Minor comments

- The abstract can be improved for a broader readership by adding some limitations, implications, and recommendations at the end of the abstract.

- Would be easier to read "dQ/dt" and change q to Q in Equation 1.

- Many papers for dQ/dt-Q-fitting are mentioned (P2L5-14), but also the key paper from Kirchner (2009) should be added here for the reader.

- Please comment a little bit more on why numerous studies have focused on event-scale recession analysis (P2L16) instead of using a classical analysis.

- Section 2.2.2 gives the impression that minimum length is a relatively new feature in recession analysis (cited papers are not older than 2012). I think this "rule" is relatively old. At least the authors can prove that the timescales of 4 and 10 days are valuable and in line with the literature.

- Please explain more detailed how the fitting for L=1 and L=0 is technically done (P6L27-29).

- Readers are referred to Dralle et.al (2015) to understand the scale-corrected recession scale parameter. As this step is again important to validate the results more explanation for this step is needed here (P7L2).

- Please clarify what is meant by "higher order moments" (P7L23).

- Where is the mentioned illustration of the patterns for the Elder Creek watershed (P8L24+25). This is confusing for the reader: "These plots provide visual representation. . .." (P8L25,26) – where?

- Only the last two sentences of the first paragraph in 3. Results are really results. Please revise this paragraph. I don't think that an explanation about Spearman rank in general is actually needed here.

- Some of the graphs are somehow blurred in the PDF (e.g. Fig2+3), please check the alpha value or increase the hue of the colors.

- Please highlight in each Âňfigure caption whether the graph is showing boxplot for all catchment or only for the example Elder Creek!

- Where is the "subplot" (P11L3)? Is the specific line in the heatmap meant here?

- Please illustrate/clarify (table or graph) the important results/discussion of section 4.2. The link to Fig 4 is a little bit weak here.

- Remove Eq.3. and explanation around the W metric from discussion to method.

- Author contributions are missing.

Fig3: Add method code legend (0000,0001,etc), remove graph title, this graph is perhaps simplified by use of boxplots, but it is not clarified. What can we learn from this graph? In my opinion this is poor graph only showing the range of MAPE for all methods and a little bit the spread of values in each boxplot. Please consider re-arranging the single boxplots (e.g. all xx10 beside each other to highlight the small MAPE for this group). There is an attempt to guide the reader through the graph (P10L4-11), but graph and text correspond not very well to each other. Please consider changing the coding for the methods, even in the text the authors mentioned that C and L subdivide the results into some groups. Please illustrate these finding also in the graph. It is worthwhile to consider an other type of visualization here (perhaps a line plot?).

Fig4: The same is true for Figure 4, are boxplots here really the best choice, I don't

think so, because it would be interesting to "follow" the a,b and Tr values of a specific catchment for all the methods. I cannot see if this catchment with the lowest a or b value for let's say method 0001 is also the lowest for other methods (e.g. 0011 or 1001). Again, it would be helpful to have a legend here with the method coding (M-S-C-L).

Fig5: Legend is "not shared", in the text it is often "unshared", please make this consistent.

Tab2. All relevant information is there, but the table is not well illustrated and hard to read. A "decision tree" or similar approaches would be

---

## Author Comment (AC1) · 7 Nov 2016

**Response to reviewer 1**

We thank the reviewer for constructive feedback and positive comments.

**Response to major comments**

**M1:** We completely agree with the reviewer; more information should be given concerning catchment features, size, and climatology. In response to this comment, we will include new wording clarifying that the study focuses on forested, relatively steep, rainfall dominated catchments, without significant snowfall and without significant regional groundwater systems. We will also include new language and a figure demonstrating the relatively erratic nature of the flow regime in the study catchments. Such a regime is ideal for recession sensitivity analysis, as the catchments "explore" a large range of recession behaviors and wetness states.

**M2:** The reviewer's comment summarizes the purpose of our work.

**M3:** We thank the reviewer for noticing this. Snow is an unimportant feature in our catchments, which are entirely rain dominated coastal watersheds. We will make this clearer in the revised manuscript, which will include more information on the features, climatology, and flow regime of the study catchments.

**Response to minor comments**

**m1:** The reviewer is correct. However, we included this citation because *Ye et al.* (2014) extract individual, contiguous periods of recession with constraints similar to those mentioned in many event-scale analyses. This is contrasted with *Brutsaert and Nieber's* (1977) proto-typical "bulk" recession analysis method, which completely avoids the issue of extracting contiguous segments.

**m2:** We are grateful for the reviewer's attention to detail. We will review our citations list and fix these issues.

---

## Author Comment (AC2) · 7 Nov 2016

**Response to reviewer 2**

We thank the reviewer for an extremely thorough review and for numerous constructive suggestions. In the following, we have addressed the reviewer's primary issues, which relate to the contextualization of the manuscript objectives and findings, manuscript ordering and organization, and visualization of data.

**Response to major comments**

**M1:** The reviewer makes two important points in this first comment:
- **Method choices:** We completely agree. While the relevance of each choice is detailed in the section *Section 2.2: Overview of the methods varied across recession analyses*, the introduction would benefit from a brief overview of these choices and their prevalence in the literature. We will include this in the revised manuscript.
- **What defines a good catchment?** We agree with the reviewer that it is important to describe the characteristics of a catchment that would be relevant to our study. We will include wording clarifying that the study focuses on forested, relatively steep, rainfall dominated catchments, without significant snowfall and without significant regional groundwater systems.

**M2:** We completely agree with the reviewer that erratic streamflow regimes would be best for this type of study. Fortunately, these study catchments, from coastal California and Oregon, exhibit highly seasonal [Fatichi et al, 2012] Mediterranean climates. Consequently, the watersheds exhibit high inter and intra-annual variability in streamflow. Quantitatively, *Botter et. al.* [2013] define erratic vs. persistent streamflow regimes using a gamma distribution fit to the streamflow empirical frequency histogram. Erratic regimes are those for which the probability distribution function (PDF) is monotonically decreasing in Q (mode at zero flow, but with a heavy tail), and persistent regimes are those for which the PDF is humped at some value of Q greater than zero. For the Eel River watershed (one of the watersheds featured in our study), we have performed many such probabilistic analyses [e.g. Dralle et. al., 2015], and most pacific coast Mediterranean watersheds are classified as erratic by this metric. To demonstrate, we present a typical year taken from the USGS gage on the Eel River watershed, at the Scotia, CA, along with the corresponding period-of-record streamflow PDF derived *only* from wet season months (Nov – April):

[Figure]

Even without including the dry season period (May – Oct) in the above PDF, the best fit gamma distribution is clearly monotonic, indicating an erratic streamflow regime.

The reviewer's comment, however, indicates that we did not clearly describe the features of the flow regimes of the study catchments, which is critical for understanding the relevance of the results. We will follow the reviewer's suggestion to include more information on the catchments' streamflow regimes, climatic features of the region, and a plot similar to the one above demonstrating the highly variable nature of the flow time series.

**M3:** We thank the reviewer for this aesthetic suggestion. The edited manuscript will include improved labels (especially in Figures 3 and 4) to make the individual method choices more clear. For more details on figure changes, see **m16.**

**M4:** We agree with the reviewer; the peak filtering approach could be described more clearly. The edited manuscript will include a new figure illustrating the peak selection algorithm, along with a sketch demonstrating the method for determining recession end.

**M5:** The reviewer makes a good point. The edited manuscript will outline these

research questions more clearly in the introduction, preparing the reader for the more complete description found in Section 2.3, which cannot reasonably be presented prior to outlining both the method combinations (MSCL) and the recession measures ($a$, $b$, $T_R$).

**M6:** This is a good question. Instead of single values for *a, b,* and *Tr*, our analyses provide populations of these variables for each catchment. However, to rank catchments, we needed single number descriptors of the population. Obvious choices could include the mean and median for measures of central tendency, and standard deviation or the inter-quartile range for variability. We did not want the occasional erroneous fit confounding our rankings, and so we chose to use the median and inter-quartile range, which are robust against biasing effect of outlier fits. We will add language in the manuscript explaining this choice.

**M7:** We agree with the reviewer; the paper could benefit from some sort of summary figure detailing the steps of analysis. We will add a decision tree to the edited manuscript, detailing the ordering of extraction and fitting, the variables derived from these procedures, and the subsequent analyses performed on the populations of recession measures.

**M8:** We thank the reviewer for this observation. We will take time during the first revision to separate any discussion points from the results section, and vice versa.

**M9:** We agree with the reviewer, this information should and will be included in the table. We will also include discussion on the potential effect of different record lengths on the results.

**M10:** We agree that the numerous box plots may be somewhat un-inspired. For full details on numerous figure changes, see **m16**.

**Response to minor comments**

**m1:** The edited abstract will include discussion of these points.

**m2:** This will be changed.

**m3:** We thank the reviewer for mentioning this important citation; it will be added.

**m4:** Thank you, this would be useful to include. We will add a few sentences mentioning the various motivations for event-scale analysis.

**m5:** There are few studies prior to the mid-2000's that extract individual

recession events. Some, such as *Wittenberg* (1999) extract individual events for non-linear fitting, but provide no methodological information concerning the minimum recession length. In the context of bulk recession analysis, the method introduced by *Brutsaert and Nieber* (1977) removed the need to identify the start/end of a recession event, and so "minimum length" is not typically discussed. Still, some studies prior to 1977 (e.g. *Howe* 1966) do extract individual events, and some mention minimum length requirements. We will cite these papers, along with some bulk recession analysis papers that include minimum length requirements.

**m6:** The fitting procedures will be more thoroughly described.

**m7:** The scale correction procedure will be more thoroughly described.

**m8:** We will clarify this (e.g. variance of $b$ is a higher order moment of the distribution of $b$), thank you.

**m9:** We thank the reviewer for catching this. We will explicitly cite the Figure number, and will also clearly delineate which figures present Elder Creek results, and which present results for all watersheds.

**m10:** We agree with the reviewer and will remove all but the last two sentences of the first paragraph in Section 3. The description of the Spearman rank will be relegated to the methods section.

**m11:** This may be related to the fact that these figures were saved as .png files. The revised manuscript will include higher resolution versions of these figure images, which should fix the blurring issue.

**m12:** We thank the reviewer and will implement this suggestion (relevant changes described in **m9**).

**m13:** We agree that this is somewhat unclear and will clarify this reference.

**m14:** We only reference Figure 4 to give the reader a sense of the range over which the recession measures ($a, b,$ and $Tr$) typically vary. The important results here are that some measures (e.g. $a$) were found to be considerably more robust with respect to ranked analysis than others (e.g. $Tr$). This has implications for comparative recession analyses, where the relative values of recession measures are used to classify or contrast catchments. We will make this clearer in our discussion.

**m15:** We will transfer this analysis to the methods section and present the plot in the results section.

**m16:** We agree with the reviewer, the plot could be re-arranged to better facilitate and match the discussion section. In order to address the reviewer's general concerns about figure quality, the following specific changes to figures 3, 4, and 5 will be implemented:

- Figure 3 will lump **00, **01, **10, **11 to make 4 plots, rather than 16 separate boxplots -- the message being that concavity and linearity choices are the primary drivers of fit quality.
- Figure 4 will display on median(b) vs. (**00, **01, **10, **11) and IQR(b) vs. (00**, 01**, 10**, 11**), since the primary finding is that median(b) increases along the sequence **00, **01, **10, **11 and IQR(b) decreases along the sequence (00**, 01**, 10**, 11**). This strategy will be used for *a* and *Tr* as well.
- Similar to figure 4 changes, figure 5 will plot only distributions of medians and distributions of IQRs for 0***, 1***.

In all figures, we will add a code legend, and will implement the reviewer's suggestion to use more intuitive labels for the methods (i.e. 0000 → mscl, or 0101 → mScL).

**m17:** We believe the reviewer misunderstood the purpose of this plot. These are exactly as the reviewer suggested: plots of *a, b,* and *Tr* across all methods for a single catchment, Elder Creek. As the reviewer suggested earlier, however, this confusion could be avoided with more clear labeling when the results are relevant to Elder Creek, or to all catchments. We intend to more clearly label plots in this way.

**m18:** We will encode this information into a decision tree and pair this with a diagram showing the steps of analysis (also see **M7**).

**Bibliography**

Fatichi, S., V. Yu Ivanov, and E. Caporali. "Investigating interannual variability of precipitation at the global scale: Is there a connection with seasonality?." *Journal of climate* 25.16 (2012): 5512-5523.

Botter, Gianluca, et al. "Resilience of river flow regimes." *Proceedings of the National Academy of Sciences* 110.32 (2013): 12925-12930.

Dralle, David N., Nathaniel J. Karst, and Sally E. Thompson. "Dry season streamflow persistence in seasonal climates." *Water Resources Research*

(2016).

Wittenberg, Hartmut. "Baseflow recession and recharge as nonlinear storage processes." *Hydrological Processes* 13.5 (1999): 715-726.

Brutsaert, Wilfried, and John L. Nieber. "Regionalized drought flow hydrographs from a mature glaciated plateau." *Water Resour. Res* 13.3 (1977): 637-643.

Howe, J. W. "Recession characteristics of Iowa streams: Part I – Temporal and areal distribution of recession constants." (1966).

---

## Author Response (AR1)

Dear Editor,

In response to the reviewers' comments and your recommendations we have implemented numerous changes to the manuscript. The primary changes include:

1) Additional climatological and physiographic information on study catchments, including a "periods of record" figure and an illustrative figure showing a typical study catchment flow regime.
2) Enumeration of four primary research questions, introduced in the introduction, around which the methods, results, and discussion are organized.
3) Two new conceptual figures designed to simplify comprehension of the numerous methods of analysis; one figure demonstrates the peak selection algorithm (figure 4), and the other presents a decision tree (figure 3; old table 2 was removed) illustrating the 4 method choices and their consequences on recession extraction.
4) Clarification edits to results figures (better labeling and organization), as well as a total redesign of the Mean Absolute Percent Error figure (new Figure 6), and the shared vs. unshared analysis (new Figure 9).
5) Reorganization of the results and discussion sections to ensure discussion topics are not mixed with results.
6) Addition of a fifth author, Andrew Veenstra of UC Berkeley, who has assisted with new analyses and with the development of recession analysis software, which is now linked in the acknowledgements.

The reviewers' comments certainly identified areas for improvement and have strengthened the paper, and we thank them for their input and suggestions. Please find below our responses to the reviewers' comments, as well as the marked-up version of the revised manuscript. We look forward to your evaluation of this revision.

Yours truly,

David Dralle

**Response to reviewer 1**

We thank the reviewer for constructive feedback and positive comments.

**Response to major comments**

**M1:** We completely agree with the reviewer; more information should be given concerning catchment features, size, and climatology. A more detailed description of the study catchments has been added to Section 2.1. The abstract also highlights that results are relevant to watersheds that are relatively steep, forested, and rain-dominated. Additionally, a new figure has been added to illustrate a representative year of seasonally dry streamflow data, from the Elder Creek catchment:

[Figure]

We also include the following figure to illustrate the periods of record of the study catchments:

[Figure]

**Figure 1.** Green lines correspond to periods during which streamflow data was available for each catchment.

**M2:** The reviewer's comment summarizes the purpose of our work.

**M3:** We thank the reviewer for noticing this. Snow is an unimportant feature in our catchments, which are entirely rain dominated coastal watersheds. We have made this clearer in the revised manuscript, which includes more information on the features, climatology, and flow regime of the study catchments.

**Response to minor comments**

**m1:** The reviewer is correct. However, we included this citation because *Ye et al.* (2014) extract individual, contiguous periods of recession with constraints similar to those mentioned in many event-scale analyses. This is contrasted with *Brutsaert and Nieber's* (1977) proto-typical "bulk" recession analysis method, which completely avoids the issue of extracting contiguous segments.

**m2:** We are grateful for the reviewer's attention to detail. We have reviewed our citations list and fixed these issues.

**Note:** Please see response to Reviewer 2 for more details concerning manuscript re-organization, as well as new figures.

**Response to reviewer 2**

We thank the reviewer for an extremely thorough review and for numerous constructive suggestions. In the following, we have addressed the reviewer's primary issues, which relate to the contextualization of the manuscript objectives and findings, manuscript ordering and organization, and visualization of data.

**Response to major comments**

**M1:** There are two major additions to the introduction:
1) We have included brief descriptions of the four method choices in an enumerated list, providing more justification for these choices.

1. *The minimum allowable length of a recession event* – This choice sets a minimum duration (units of days in our analysis) for a recession period to be selected for analysis. Recessions less than the minimum duration are discarded.

30    2. *The definition of the beginning of a recession event* – The start of a recession event is usually determined by a flow peak filtering algorithm applied to the streamflow time series. Commonly, peaks are identified using a simple flow threshold, wherein flow peaks exceeding the threshold are flagged as potential starts to a recession event.

3. *The definition of the end of a recession event* – Recession ends can be identified by the occurrence of a rainfall event, a transition from decreasing discharge to increasing discharge ($dQ/dt < 0 \rightarrow dQ/dt > 0$), or a break in the upward concavity of the flow time series ($d^2Q/dt^2 > 0 \rightarrow d^2Q/dt^2 < 0$), among other criteria.

4. *The method of power law model fitting* – Numerous methods for fitting the power law recession model have been de-
5    veloped. Most such methods involve either some form of linear regression on the log transformed version of Equation 1 ($\log[-dQ/dt] = \log a + b \log Q$), or nonlinear regression on the solution to Equation 1

On a related note, the start of Section 2.2. defends the four method choices as the most "fundamental" choices that one must make for event-scale analysis:

**2.2   Overview of the methods varied across recession analyses**

Presumably there is an unbounded range of methodological choices that could be made regarding event-scale recession analy-
25   sis. To constrain the problem, we address, in the simplest manner possible, the decisions that all analyses must confront: (i) the selection of a minimum duration of time for any candidate recession, (ii) the selection of a time point signifying the recession start (peak selection), (iii) the selection of criteria to confirm the continuation of a hydrograph segment that merits analysis (e.g. a slope or concavity requirement), (iv) the selection of a fitting methodology by which to analyze a chosen recession. While other choices undoubtedly have impacts on the characteristics of a population of analyzed recessions, the selection of
30   these four criteria represent the most constrained and fundamental set of methodological choices to explore.

2) We have introduced a list of four primary research questions, around which the Results and Discussion sections have been re-organized:

*Research question 1* – How do methodological choices impact fit quality of the power law recession model?

*Research question 2* – When catchments are ranked by fitted recession parameter statistics, is the rank order dependent on methodological choices?

*Research question 3* – How do methodological choices affect the empirical frequency distributions (over the period of record) of recession parameter values?

*Research question 4* – How might methodological choices affect relationships between a given recession parameter and other physical measures of catchment state, such as catchment wetness?

**M2:** A more detailed description of the study catchments has been added to Section 2.1. The abstract also highlights that results are relevant to watersheds that are relatively steep, forested, and rain-dominated. Additionally, a new figure has been added to illustrate a representative year of seasonally dry streamflow data, from the Elder Creek catchment:

[Figure]

**M3:** We have drastically changed figure presentation so that the MSCL coding is rarely referenced. In the one case that it is mentioned (Figure 8), we now include MSCL letters under the binary codes to help readers identify this correspondence. Please see **m17** for figure details.

**M4:** We agree with the reviewer; the approach could be more clearly illustrated. We have included a new figure, which corresponds to the text description:

[Figure]

**Figure 4.** Illustration of the peak extraction algorithm. The square represents the most recent recession peak identified for selection. The empty star identifies a local maximum that will not be selected due to the fact that the subsequent recession does not decay by an amount $X$ before the next local maximum. The filled star is selected as a recession peak because it is at least $X$ greater than the local minimum between it and the previously selected recession peak (square), and is followed by a flow decrease of at least $X$.

The "concavity" and "decreasing" recession-end concepts are also illustrated in the new decision tree (Figure 3), detailed below in **M7.**

**M5:** The reviewer makes a good point. The edited manuscript now includes an outline of these research questions in the introduction. A fourth research question (addressing the antecedent wetness exercise) has been added as well; see comment **m15**.

**M6:** This is a good question. Instead of single values for *a, b,* and *Tr*, our analyses provide populations of these variables for each catchment. However, to rank catchments, we needed single number descriptors of the population. Obvious choices could include the mean and median for measures of central tendency, and standard deviation or the inter-quartile range for variability. We did not want the occasional erroneous fit confounding our rankings, and so we chose to use the median and inter-quartile range, which are robust against biasing effect of outlier fits. We have added language in the manuscript explaining this choice.

**M7:** We agree with the reviewer; the paper could benefit from some sort of summary figure detailing the steps of analysis. We now include a "decision tree" (Figure 3) illustrating the four method choices, along with their consequences:

[Figure]

**Figure 3.** Graphical illustration of the sixteen method choices. Minimum recession length (M) determines whether extracted recessions are required to have a minimum length of either 4 (M = 0) or 10 days (M = 1). Recession peak selectivity (S) determines whether the peak selection algorithm is highly restrictive (S = 1) or relatively permissive (S = 0). Recession concavity (C) determines whether recessions are required to be both decreasing *and* concave up (C = 1), or simply decreasing (C = 0). Finally, Linearity (L) determines whether or not the values of the recession parameters $a$ and $b$ are determined using linear regression on the plot of $\log\left[-dQ/dt\right]$ vs. $\log Q$ (L = 1), or using a nonlinear least squares fit to the raw recession time series (L = 0).

The shared vs. unshared analysis is still illustrated in Figure 5. Please see comment **m17** for an updated figure that more clearly illustrates the connection between the shared vs. unshared comparisons and the Mann Whitney U test.

**M8:** We thank the reviewer for this observation. We have thoroughly re-organized the methods/results/discussion to clarify the content of these three sections. Additionally, the sections are now more tightly organized around the research questions introduced in the introduction.

**M9:** We agree with the reviewer. To address this comment, we have first introduced a new figure showing the period of record of all catchments:

[Figure]

**Figure 1.** Green lines correspond to periods during which streamflow data was available for each catchment.

Additionally, we performed some very basic stationarity tests to ensure results are not sensitive to the period of record. The following text has been introduced to the start of the Results section:

**3 Results**

While the lengths of record for study catchments vary from 35 - 99 years, we find that subsetting flow records and re-performing analysis does not significantly impact our findings. We also find that, at confidence level $p = 0.05$, approximately 6% of the (16 catchments $\times$ 16 method combinations $\times$ 3 recession measures) $= 768$ populations of recession measures exhibit significant trends over time. At a confidence level of 0.05, one would expect 5% of the tests to flag significance purely by chance. We conclude that any potential trends in recession parameters over time will have a minimal impact on the results of this study.

**M10:** We agree that the numerous box plots may be somewhat un-inspired. For full details on numerous figure changes, see **m17**.

**Response to minor comments**

**m1:** The edited abstract includes recommendations, and limitations based on study catchment features.

**m2:** This has been changed.

**m3:** We thank the reviewer for mentioning this important citation; it has been added.

**m4:** Thank you, this would be useful to include. We have included new text addressing motivations of event-scale analysis:

catchment state (e.g. Biswal and Marani, 2010; Shaw and Riha, 2012). Motivations for event-scale analysis include testing physical theories that predict variability in power law streamflow recessions (e.g Harman et al., 2009; Biswal and Marani, 2010), detection of human land use impacts on catchment water balance (e.g. Bogaart et al., 2016), and prediction of extent of the wetted channel network (Ghosh et al., 2016; Shaw, 2016).

**m5:** *Howe* (1966) has been cited as an older example of a manuscript which includes a 10 day minimum recession length. We also point out that most lumped recession analyses do not choose minimum recession lengths, owing to the derivative based method of Brutsaert and Nieber (1977).

**2.2.2 Defining the minimum allowable length of recession event (M)**

Owing to the derivative based methods developed by Brutsaert and Nieber (1977), most lumped recession analyses do not set a minimum duration recession events. However, nearly all event-scale recession studies set a minimum duration for chosen recession periods. Reasons for this choice vary; authors cite the removal of noise from short events (Ye et al., 2014), the necessity of capturing late time flow processes (Chen and Krajewski, 2015), and data quality concerns related to sample size (Shaw, 2016). Event-scale recession analyses have typically chosen a minimum of 4 to 5 days of recession for daily data (e.g. Shaw and Riha, 2012; Biswal and Marani, 2010), although values upwards of 10 days (e.g. Howe, 1966) and as low as 12 hours (e.g. McMillan et al., 2014, for high frequency data) have been used.

**m6:** The fitting procedures have been more thoroughly described:

For the purposes of the present study, we again frame the problem in terms of the most fundamental methodological dichotomy between linear and nonlinear fitting. Linear fitting (L=1) is performed on the log-transformed values, $[\log{(Q)}, \log{(-dQ/dt)}]$. Values of the flow derivative are computed for each two day window (days $i$ and $i-1$, with $\Delta t = 1$ day) over the duration of the recession as $dQ/dt = (Q_i - Q_{i-1})/\Delta t$, with corresponding values of $Q$ computed as the average flow value over both days, $Q = (Q_i + Q_{i-1})/2$ (Brutsaert and Nieber, 1977). Nonlinear fitting (L=0) is performed using nonlinear least squares minimization on extracted, non-transformed recession segments.

**m7:** The scale correction procedure has been more thoroughly described:

The scale correction procedure begins by first fitting each recession curve to obtain an initial population (of size $n$) of recession parameters $a_i$ and $b_i$. The flow time series is then re-scaled by a constant, $Q_0$, computed as:

$$Q_0 = \exp\left[\frac{\sum_{i=1}^{n}(b_i - \bar{b})(\log a_i - \overline{\log a})}{\sum_{i=1}^{n}(b_i - \bar{b})^2}\right],\tag{2}$$

where $\overline{\log a}$ is the mean of the natural logarithm of the $a_i$, and $\bar{b}$ is the mean of the $b_i$. Following re-scaling, the flow time series
20    is re-fit to the power law recession model. While the recession exponent is scale-independent, the recession scale parameter will be altered by the scaling procedure in such a way as to eliminate artifactual linear correlation between $\log a$ and $b$. The resultant population of recession scale parameters has units of inverse time and has been shown empirically to correlate strongly with measures of catchment wetness (Dralle et al., 2015).

**m8:** We have clarified this.

**m9:** We have clarified this.

**m10:** We agree with the reviewer and have removed all but the last two sentences of the first paragraph in Section 3. The description of the Spearman rank has been relegated to the methods section.

**m11:** New figures have been added. See **m17** for an overview.

**m12:** We clearly label figures that present Elder Creek results.

**m13:** We have clarified this reference.

**m14:** The reference to Figure 4 has been removed. The important results here are that some measures (e.g. *a* ) were found to be considerably more robust with respect to ranked analysis than others (e.g. *Tr*). This has implications for comparative recession analyses, where the relative values of recession measures are used to classify or contrast catchments. We have made this clearer in the discussion.

**m15:** We have transferred this analysis to the methods section; it is now used to address the "fourth" research question added to the introduction.

**m16:** Contributions have been added.

**m17:** We agree with the reviewer; most plots could be re-arranged to better facilitate and match the discussion section. The following new/updated figures have been added:

1) New Figure 6 (MAPE plot): Now lumped by concavity and linearity (as the reviewer suggested), the two important method choices highlighted in the discussion. Given that the other method choices were not found to be important, this greatly simplifies the presentation. Additionally, all binary references to the methods were removed and the actual method choices are labeled:

[Figure]

**Figure 6.** Mean absolute percentage error (MAPE) lumped across catchments by three groups: Concave only recessions with nonlinear fitting; Concave recessions or nonlinear fitting but not both (denoted using the logical 'exclusive or' operator, 'xor'); and decreasing recessions (without the concavity requirement) and linear fitting procedures.

2) New Figure 8 (recession measure distributions for Elder Creek): We have retained the plot of all recession measures across all method combinations for Elder Creek. We believe at least one plot should illustrate the various effects of all 16 method combinations. However, we have added new

labeling to identify these results as relevant to the Elder Creek catchment, and have added labeling to the horizontal axes to remind the reader of the correspondences between the method choices and the binary codes.

3) New Figure 9 (shared vs. unshared distributions for Elder Creek): To clarify the connection between the Mann-Whitney U test and the shared vs. unshared distributions, we simplified and edited the shared vs. unshared distribution figure. First, we removed the binary codes referencing the 8 combinations of the other method choices, as it is not an important detail for the analysis. Additionally, we only present shared vs. unshared distributions for two method choices: concavity and linearity. Finally, we highlight which distributions are identified as significantly different according to the Mann Whitney U test.

[Figure]

**Figure 9.** Box plots comparing recession exponent shared vs. unshared distributions for minimum recession length and concavity method choices for Elder Creek. Each sub-plot corresponds to a particular method choice; the shared boxes are generated with the $b$ values from the recessions shared between the more and less restrictive values of the method choice for that sub-plot. The unshared boxes are those values of $b$ from the recessions extracted by *only* the less restrictive value of the subplot method choice. The independent axis enumerates the eight combinations of the method choices other than the subplot method choice.

4) Thanks to the reviewer's comments concerning the varying periods of record for study catchments, we slightly altered the Mann Whitney sensitivity plot. We realized that differing record lengths between catchments may lead to discrepancies in the sample sizes between rows of the sensitivity plot. Therefore, we plot the *ranking* of sensitivities for each row, instead of the fraction of Mann Whitney U tests that identify significant differences between the shared and unshared distributions. This improves comparability between rows:

[Figure]

**Figure 10.** Results of Mann-Whitney U test sensitivity analysis. Each row represents one of the 16 study catchments, each subplot one of the three recession measures $a$, $b$, or $T_R$, and each subplot column one of the four methodological choices (MSCL). Each cell is colored by a sensitivity rank. A cell shading of 4 (darkest) means that method choice had the highest number of significantly different shared and unshared distributions for that recession measure in that catchment, indicating that the particular recession measure is highly sensitive to the corresponding method choice.

5) As the reviewer suggested, we removed Table 2 and replaced it with a decision tree (see **M7**).

*Correspondence to:* David N. Dralle (dralle@berkeley.edu)

**Abstract.**

The study of single streamflow recession events is receiving increasing attention following the presentation of novel theoretical explanations for the emergence of power-law forms of the recession relationship, and drivers of its variability. Individually characterizing streamflow recessions often involves describing the similarities and differences between model parameters fitted to each recession time series. Significant methodological sensitivity has been identified in the fitting and parameterization of models that describe populations of many recessions, but the dependence of estimated model parameters on methodological choices has not been evaluated for event-by-event forms of analysis. Here, we use daily streamflow data from 16 catchments in northern California and southern Oregon to investigate how combinations of commonly used streamflow recession definitions and fitting techniques impact parameter estimates of a widely-used power-law recession model. Results are relevant to watersheds that are relatively steep, forested, and rain-dominated. The highly seasonal mediterranean climate of northern California and southern Oregon ensures study catchments explore a wide range of recession behaviors and wetness states, ideal for a sensitivity analysis. In such catchments, we show that: (i) methodological decisions, including ones that have received little attention in the literature, can impact parameter value estimates and model goodness-of-fit; (ii) the central tendencies of event-scale recession parameter probability distributions are largely robust to methodological choices, in the sense that differing methods rank catchments similarly according to the medians of these distributions; (iii) recession parameter distributions are method-dependent, but roughly catchment-independent, such that changing the choices made about a particular method affects a given parameter in similar ways across most catchments; and (iv) the observed correlative relationship between the power law recession scale parameter and catchment antecedent wetness varies depending on recession definition and fitting choices. Considering study results, we recommend a combination of four key methodological decisions to maximize the quality of fitted recession curves, and to minimize bias in the related populations of fitted recession parameters.

**1   Introduction**

Streamflow recession analysis has the goal of characterizing recession behavior in terms of phenomenological models of decreases in flow ($Q$, with units of [L T$^{-1}$] or [L$^3$ T$^{-1}$]) over time, typically represented with a power-law differential equation (Boussinesq, 1877; Hall, 1968; Tallaksen, 1995):

$$\quad \underset{\sim}{\frac{dq}{dt}}\frac{dQ}{dt} = \underline{-aq}-aQ^b \implies \underset{\sim}{q}Q(t) = \left(\underset{\sim}{q}
[revised manuscript text omitted]

[Figure]

**Example shared vs. unshared distributions for Elder Creek watershed**

Red highlight means the Mann-Whitney U test detected a significant difference between shared and unshared distributions.

**Figure 9.** Box plots comparing  recession exponent shared vs. unshared distributions for  minimum recession  length and concavity method choices for Elder Creek. Each sub-plot corresponds to a particular method choice; the shared boxes are generated with the $b$ values from the recessions shared between the  more and  less restrictive values of the  method choice for that sub-plot. The unshared boxes are those values of $b$ from the recessions extracted by  *only* the less restrictive value of the subplot method choice. The independent axis  enumerates the  eight combinations of the method choices other than the subplot method choice.

[Figure]

**Figure 10.** Results of Mann-Whitney U test sensitivity analysis. Each row represents one of the 16 study catchments, each subplot one of the three recession measures $a$, $b$, or $T_R$, and each subplot column one of the four methodological choices (MSCL). Each cell is colored by a sensitivity rank. A cell shading of 4 (darkest) means that method choice had the highest number of significantly different shared and unshared distributions for that recession measure in that catchment, indicating that the particular recession measure is highly sensitive to the corresponding method choice.

[Figure]

**Figure 11.** The recession scale parameter plotted against antecedent catchment wetness for three method combinations, together with a linear fit for each point cloud, and a 95% confidence interval for each fitted slope.